

# Prevalence of near-death experiences in people with and without REM sleep intrusion

Daniel Kondziella[1,2,3], Jens P. Dreier[4,5,6,7,8] and Markus Harboe Olsen[9]

[1] Department of Neurology, Rigshospitalet, University of Copenhagen, Copenhagen, Denmark

[2] Faculty of Health Sciences and Medicine, University of Copenhagen, Copenhagen, Denmark

[3] Department of Neuroscience, Norwegian University of Technology and Science, Trondheim, Norway

[4] Center for Stroke Research Berlin, Charité—Universitätsmedizin Berlin, corporate member of Freie Universität Berlin, Humboldt-Universität zu Berlin, and Berlin Institute of Health, Berlin, Germany

[5] Department of Neurology, Charité—Universitätsmedizin Berlin, corporate member of Freie Universität Berlin, Humboldt-Universität zu Berlin, and Berlin Institute of Health, Berlin, Germany

[6] Department of Experimental Neurology, Charité –Universitätsmedizin Berlin, corporate member of Freie Universität Berlin, Humboldt-Universität zu Berlin, and Berlin Institute of Health, Berlin, Germany

[7] Bernstein Center for Computational Neuroscience Berlin, Berlin, Germany

[8] Einstein Center for Neurosciences Berlin, Berlin, Germany

[9] Department of Neuroanesthesiology, Rigshospitalet, University of Copenhagen, Copenhagen, Denmark

Corresponding author
Daniel Kondziella,
daniel_kondziella@yahoo.com

## ABSTRACT

**Background.** The origin and prevalence of near-death experiences are unknown. A recent study suggested a link with REM sleep intrusion but was criticized for its selection of control participants. We therefore assessed the association of REM intrusion and near-death experiences with different methods.

**Methods.** Using a crowd-sourcing platform, we recruited 1,034 lay people from 35 countries to investigate the prevalence of near-death experiences and self-reported REM sleep intrusion. Reports were validated using the Greyson Near-Death Experiences Scale (GNDES) with ≥7 points as cut-off for near-death experiences.

**Results.** Near-death experiences were reported by 106 of 1,034 participants (10%; 95% CI [8.5–12%]). Evidence of REM intrusion was more common in people with near-death experiences ($n = 50/106$; 47%) than in people with experiences with 6 points or less on the GNDES ($n = 47/183$; 26%) or in those without such experiences ($n = 107/744$; 14%; $p = <0.0001$). Following multivariate regression analysis to adjust for age, gender, place of residence, employment and perceived danger, this association remained highly significant; people with REM intrusion were more likely to exhibit near-death experiences than those without (OR 2.85; 95% CI [1.68–4.88]; $p = 0.0001$).

**Discussion.** Using a crowd-sourcing approach, we found a prevalence of near-death experiences of 10%. While age, gender, place of residence, employment status and perceived threat do not seem to influence the prevalence of near-death experiences, we confirmed a possible association with REM sleep intrusion.

# INTRODUCTION

Near-death experiences can be defined as conscious perceptual experiences, including emotional, self-related, spiritual and mystical experiences, occurring in a person close to death or in situations of imminent physical or emotional threat (*Greyson, 1983*). Reports of near-death experiences include, but are not limited to, increased speed of thoughts, distortion of time perception, out-of-body experiences, and visual and auditive hallucinations (*Greyson, 1983*; *Knoblauch, Schmied & Schnettler, 2001*; *Van Lommel et al., 2001*; *Martial et al., 2017*; *Cassol et al., 2018*). Yet the pathophysiological basis of near-death experiences remains unknown (*Peinkhofer, Dreier & Kondziella, 2019*).

Near-death experiences share phenotypical features with those made during rapid eye movement (REM) sleep (Table 1) (*Nelson et al., 2006*). REM sleep is defined by rapid and random saccadic eye movements, loss of muscle tone, a propensity towards vivid dreams, and cortical activation as revealed by EEG desynchronization (*Peever & Fuller, 2017*). Importantly, REM state features can intrude into wakefulness, both in healthy people and those with narcolepsy, which may lead to visual and auditory hallucinations at sleep onset (hypnagogic) or upon awakening (hypnopompic) and muscle atonia with sleep paralysis and cataplexy (*Scammell, 2015*; *Jalal & Ramachandran, 2017*; *Baird et al., 2018*).

In a recent case-control study, people with near-death experiences reported significantly more often REM intrusion than age- and sex-matched controls, and the authors suggested that REM sleep intrusion might contribute to near-death experiences (*Nelson et al., 2006*). However, the study was criticized for several reasons, including selection bias (*Long & Janice Miner, 2007*). From a registry of 446 self-reported near-death experiences, 55 North American participants responded to email inquiry, fulfilled inclusion criteria and were enrolled; in contrast, controls were recruited from medical center personnel or their contacts (*Nelson et al., 2006*). Critics pointed out that controls might have been influenced by their medical background and that "an ideal comparison group would have been made up of people who had been through life-threatening events comparable to [those with near-death experiences], who matched [those with near-death experiences] on other identifying aspects such as culture; and who were willing to report their experiences on a public website, but who had not had [a near-death experience]" (*Long & Janice Miner, 2007*).

We therefore examined the association between REM sleep intrusion and near-death experiences using a different approach. We recruited a large multinational sample of unprimed lay people from a crowdsourcing platform (without any other selection criteria than English language and age ≥18 years) to estimate the prevalence of near-death experiences and to test the hypothesis that near-death experiences are associated with a propensity for REM sleep intrusion.

## MATERIALS & METHODS

### Hypotheses and research questions

The objectives of this study were two-fold.

- Primary objective: To estimate the frequency of near-death experiences and REM sleep intrusions reported in a large sample of adult humans collected from an online crowdsourcing service.
- Secondary objective: To test the hypothesis that people who report a near-death experience have a greater frequency of REM sleep intrusions.

### Study design

An online platform, Prolific Academic (https://prolific.ac/), was used to recruit a large global sample of lay people. Prolific Academic is a crowdsourcing online platform to recruit human subjects for research that compares favorably in terms of honesty and diversity of participants and data quality (*Woods et al., 2015*; *Peer et al., 2017*). Participants were recruited without any filters except for English language and age ≥18 years.

The survey was announced under the headline "Survey on Near-Death Experiences and Related Experiences". Prior to the start of the survey, participants were instructed that their monetary reward was fixed, regardless of whether they would report having had a near-death experience or not. No further information was given. We then asked participants to complete a questionnaire comprising demographic information on age, gender, employment status and place of residence, followed by four questions about REM sleep intrusion (Table 1). The questions about REM sleep intrusion were identical to those used by Nelson et al. in their case-control study (*Nelson et al., 2006*). Participants were then asked if they ever had had a near-death experience (regardless if this experience had occurred in life-threatening or non-life-threatening situations). If not, the survey ended here; if yes, we continued to inquire about this experience in detail, including all 16 items of the Greyson Near-Death Experience Scale (GNDES), the most widely used standardized tool to identify, confirm and characterize near-death experiences in research (*Greyson, 1983*). In addition, we inquired about unpleasant feelings (Table 1), which is not covered by the GNDES. Further, participants were given the possibility to describe their experience in their own words. See Table 1 for details.

### Statistics

We estimated the number of participants required to be 384, using a very high population size (300,000,000), a confidence level of 95% and a margin of error of 5%. However, as previous studies have estimated that 4–8% of the population have had a near-death experience (*Knoblauch, Schmied & Schnettler, 2001*; *Perera & Padmasekara, 2005*; *Facco & Agrillo, 2012*), we decided to enroll approximately 1,000 participants to identify an estimated 50–100 individuals with near-death experiences.

Data were analyzed using R (*R Core Team, 2018*). Categorical variables were analyzed using a chi-squared test. Continuous variables were compared using Student's t-tests. We calculated odds ratios for having near-death experiences with or without co-occurrence of REM sleep intrusion and performed multivariate logistic regression analysis to correct

**Table 1  Questionnaire on REM sleep intrusion and near-death experiences.** REM, rapid eye movements; * in contrast to the Near-Death Experience Scale, we also inquired about unpleasant experiences.

*Questions about REM sleep intrusion (1 point for each positive answer; based on Nelson et al. (2006)*

- Just before falling asleep or just after awakening, have you ever seen objects, things or people that others can't see?
- Just before falling asleep or just after awakening, have you ever heard voices, music or sounds that other people can't hear?
- Have you ever awakened and felt paralyzed or found that you were unable to move?
- Have you ever had abrupt muscle weakness in your legs or knee buckling, or felt sudden muscle weakness in your face or head drop?

*Questions about near-death experiences*

- Near-death experiences can be defined as any conscious perceptual experience, including emotional, self-related, spiritual and/or mystical experiences, occurring in a person close to death or in situations of intense physical or emotional danger. In plain language—near-death experiences are exceptional experiences that you may have when you are dying or feel as if you were dying. Have you ever had such a near-death experience—either during a true life-threatening event or an event that just felt so?
- Was your near-death experience associated with a true life-threatening event or an event that was not life-threatening but felt so?
- In which situation did you have a near-death experience?
- Have you had more than 1 near-death experience?
- If you wish, please describe your experience as detailed as you can (optional). We are interested to know what you felt and how your experience unfolded over time.

*Greyson Near-Death Experience Scale (0–2 points for each answer; based on Greyson (1983)*

- Did time seem to speed up or slow down?
- Were your thoughts speeded up?
- Did scenes from your past come back to you?
- Did you suddenly seem to understand everything?
- Did you have a feeling of peace or pleasantness? *
- Did you have a feeling of joy?
- Did you feel a sense of harmony or unity with the universe?
- Did you see, or feel surrounded by, a brilliant light?
- Were your senses more vivid than usual?
- Did you seem to be aware of things going on elsewhere, as if by extrasensory perception or telepathy?
- Did scenes from the future come to you?
- Did you feel separated from your body?
- Did you seem to enter some other, unearthly world?
- Did you seem to encounter a mystical being or presence or hear an unidentifiable voice?
- Did you see deceased or religious spirits?
- Did you come to a border or point of no return?

for age, gender, employment status, place of residence, and whether the situation in which an experience was made was perceived as life-threatening or not. To adjust for multiple testing, we used Bonferroni correction and set the alpha level at 0.01.

To prevent HARKing (Hypothesizing After the Results are Known) (*Fraser et al., 2018*), we pre-registered the study, including all objectives, with the Open Science Framework (https://osf.io/ykr3g).

### Ethics

Participants gave consent for publication of their (anonymous) data. Participation was anonymous, voluntary and restricted to those $\geq 18$ years. Participants received a monetary reward upon completion of the survey, following the platform's *ethical rewards* principle ($\geq$ \$6.50/h). The Ethics Committee of the Capital Region of Denmark waives approval for online surveys (section 14 (1) of the Committee Act. 2; http://www.nvk.dk/english).

### Data availability statement

All de-identified raw data are provided in the Supplemental Information. The data analysis plan can be accessed via the Open Science Framework (https://osf.io/ykr3g).

# RESULTS

We recruited 1,034 lay people from 35 countries (mean age 32.7 years, standard deviation 11.3 years; 59% female; 79% fully or part-time employed or in training), most of which were residing in Europe and North America. Table 2 and Fig. 1 provide epidemiological information.

### Near-death experiences: prevalence and semiology

Two-hundred eighty-nine participants (28%; 95% CI [25–31%]) claimed to have had a near-death experience, 106 of which reached the threshold of 7 points or more on the GNDES (37%; 95% CI [31–43%]). Thus, confirmed near-death experiences were reported by 106/1,034 participants (10%; 95% CI [8.5–12%]) (Fig. 2).

Participants perceived the situation in which they had their experience slightly more often as truly life-threatening ($n = 158$; 55%) than not truly life-threatening ($n = 131$; 45%), and this was irrespective of whether their experience met criteria for a near-death experience according to the GNDES or not ($p = 0.55$; Table 3).

Near-death experiences and experiences with $\leq 6$ points on the GNDES occurred in the following situations, listed with decreasing frequency: motor accident 27% ($n = 77$), near-drowning 19% ($n = 56$); intense grief or anxiety 18% ($n = 51$), substance abuse 11% ($n = 33$); psychological distress without organic disease 9.7% ($n = 28$); physical violence other than combat 8.3% ($n = 24$), critical illness 8.0% ($n = 23$); childbirth complication 8.0% ($n = 23$); suicide attempt 6.9% ($n = 20$); anesthesia/medical procedure 6.9% ($n = 20$); cardiac arrest/heart attack 5.5% ($n = 16$); meditation or prayer 5.2% ($n = 15$); anaphylactic reaction 4.8% ($n = 14$); combat situation 3.8% ($n = 11$); syncope 1.7% ($n = 5$); epileptic seizure 1.4% ($n = 4$); and other 19% ($n = 56$).

**Table 2  Demographic data and prevalence of REM sleep intrusion.** To adjust for multiple testing, the alpha level was set to 0.01. Significant $p$ values are shown in bold script. N, number of participants; NDE, near-death experiences; REM, rapid eye movements; SD, standard deviation; *when comparing "No NDE" ($n = 744$) with confirmed near-death experiences with a Greyson NDE Scale score $\geq 7$ ($n = 106$; see Table 3), this significance is lost ($p$-value = 0.256).

| | All ($n = 1,034$) | No NDE ($n = 744$) | All claimed NDE ($n = 289$) | $p$-value |
|---|---|---|---|---|
| **Age**—mean $\pm$ SD | $32.7 \pm 11.3$ | $33.4 \pm 11.3$ | $30.8 \pm 11.1$ | **0.0006*** |
| **Gender**—$n$ (%) | | | | **0.002** |
| Female | 607 (59%) | 461 (62%) | 145 (50%) | |
| Male | 424 (41%) | 282 (38%) | 142 (49%) | |
| Other | 3 (0.3%) | 1 (0.1%) | 2 (0.7%) | |
| **Continent**—$n$ (%) | | | | 0.03 |
| Americas | 36 (3.5%) | 19 (2.6%) | 17 (5.9%) | |
| Asia | 13 (1.3%) | 8 (1.1%) | 4 (1.4%) | |
| Europe | 938 (91%) | 687 (92%) | 251 (87%) | |
| Oceania | 19 (1.8%) | 14 (1.9%) | 5 (1.7%) | |
| Other | 28 (2.7%) | 16 (2.2%) | 12 (4.2%) | |
| **Work**—$n$ (%) | | | | 0.34 |
| Full-Time | 393 (38%) | 281 (38%) | 112 (39%) | |
| Job seeking | 57 (5.5%) | 41 (5.5%) | 15 (5.2%) | |
| Not in paid work | 99 (9.6%) | 75 (10%) | 24 (8.3%) | |
| Part-Time | 217 (21%) | 166 (22%) | 51 (18%) | |
| Student | 174 (17%) | 116 (16%) | 58 (20%) | |
| Other | 94 (9.1%) | 65 (8.7%) | 29 (10%) | |
| **REM intrusion**—$n$ (%) | | | | **<0.0001** |
| $\leq 2$ criteria | 829 (80%) | 637 (86%) | 192 (66%) | |
| $\geq 3$ criteria | 204 (20%) | 107 (14%) | 97 (34%) | |

The most often reported symptoms were abnormal time perception (faster or slower than normal; reported by 252 participants; 87%); exceptional speed of thoughts ($n = 189$; 65%); exceptional vivid senses ($n = 182$; 63%); and feeling separated from one's body, including out-of-body experiences ($n = 152$; 53%).

Experiences that qualified as a true near-death experience according to the GNDES were perceived much more often as pleasant ($n = 41$; 53%) than experiences that did not ($n = 21$; 14%; $p < 0.0001$; chi-squared test; neutral experience excluded; Table 3).

Around one third of participants reporting a near-death experience or near-death-like experience stated having had two or three such experiences ($n = 92$, 32%); and some even claimed to have had more than three ($n = 10$; 3.5%).

### Near-death experiences and evidence for REM sleep intrusion

Evidence for REM sleep intrusion (i.e., $\geq 3$ criteria fulfilled) was much more common in people with experiences above the cut-off point of the GNDES ($n = 50/106$; 47%) than in people with experiences below this threshold ($n = 47/183$; 26%) or in those without any such experience ($n = 107/744$; 14%; $p = < 0.0001$; chi-squared test; Tables 2 and 3). Following multivariate regression analysis to adjust for age, gender, place of origin,

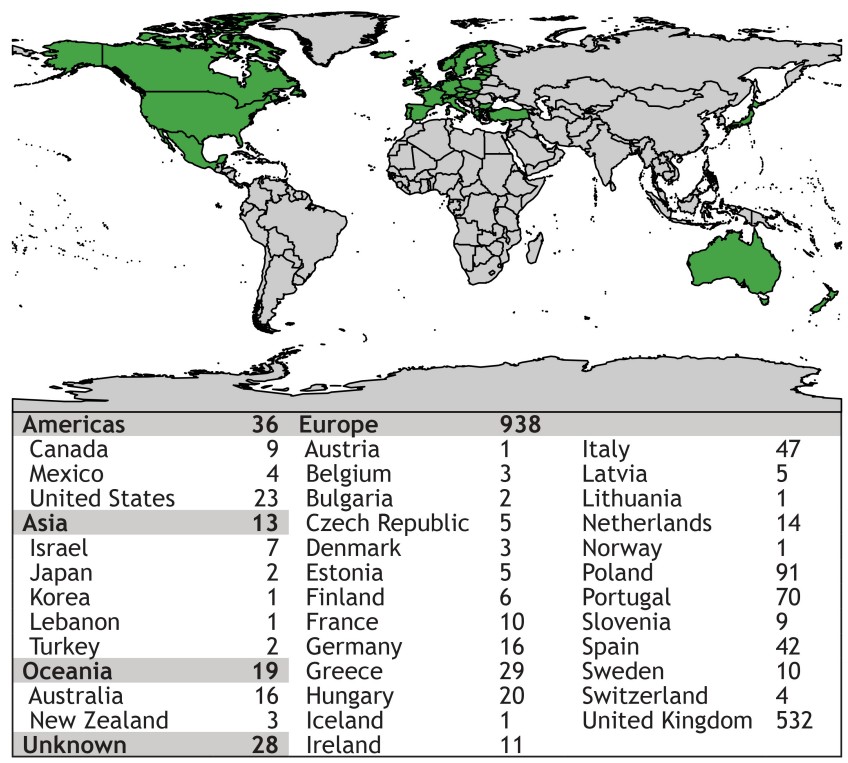

| Americas | 36 | Europe | 938 | | |
|---|---|---|---|---|---|
| Canada | 9 | Austria | 1 | Italy | 47 |
| Mexico | 4 | Belgium | 3 | Latvia | 5 |
| United States | 23 | Bulgaria | 2 | Lithuania | 1 |
| **Asia** | **13** | Czech Republic | 5 | Netherlands | 14 |
| Israel | 7 | Denmark | 3 | Norway | 1 |
| Japan | 2 | Estonia | 5 | Poland | 91 |
| Korea | 1 | Finland | 6 | Portugal | 70 |
| Lebanon | 1 | France | 10 | Slovenia | 9 |
| Turkey | 2 | Germany | 16 | Spain | 42 |
| **Oceania** | **19** | Greece | 29 | Sweden | 10 |
| Australia | 16 | Hungary | 20 | Switzerland | 4 |
| New Zealand | 3 | Iceland | 1 | United Kingdom | 532 |
| **Unknown** | **28** | Ireland | 11 | | |

**Figure 1** **Map showing places of residency of survey participants.** Using an online crowdsourcing platform, we recruited 1.034 lay people from 35 countries on four continents, the majority from Europe and North America.

employment status and perceived danger, this association remained highly significant; i.e., people with REM sleep intrusion were more likely to exhibit near-death experiences than those without REM sleep abnormalities (odds ratio 2.85; 95% CI [1.68–4.88]; $p = 0.0001$; Table 4).

Selected written reports from participants can be found in Tables 5 and 6. Raw data are provided in the *online supplemental files*.

## DISCUSSION

### Prevalence and semiology of near-death experiences

Using crowdsourcing methods, we found that one out of 10 people from a large sample of 35 countries had a confirmed near-death experience (10%; 95% CI [8.5–12%]). This estimate is slightly higher than what was reported in previous studies using traditional interview-based surveys in Australia (8%) (*Perera & Padmasekara, 2005*) and Germany (4%) (*Knoblauch, Schmied & Schnettler, 2001*); but, of note, none of those studies validated reports with the GNDES. Experiences that did not fulfill criteria for a near-death experience were roughly twice as common in our survey.

Similar to previous reports, we found that near-death experiences occur in various cultures and nationalities and irrespective of employment status, age and gender
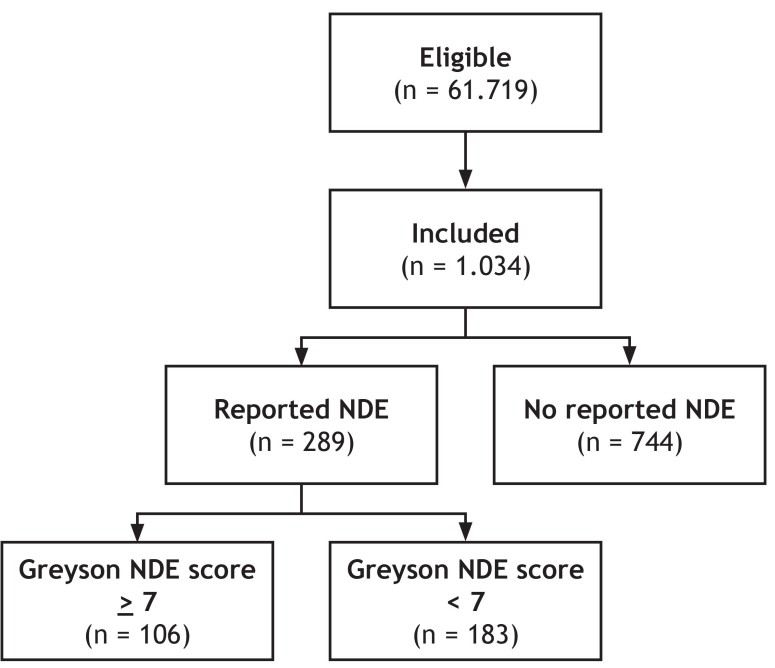

**Figure 2** **Schematic overview of study design.** Of 61.719 eligible lay people registered with Prolific Academic (https://prolific.ac/; accessed on January 22, 2019), we enrolled 1.034 participants; 106 (10%; CI 95% 8.5–12%) of whom reported a near-death experience that fulfilled established criteria (Greyson Near-Death Experience Scale score of 7 or higher). N, number of participants; NDE, near-death experience.

(*Thonnard et al., 2013*; *Charland-Verville et al., 2015*; *Martial et al., 2017*; *Martial et al., 2018*; *Cassol et al., 2018*). However, unlike most previous reports in which near-death experiences were almost always associated with peacefulness and well-being (*Thonnard et al., 2013*; *Charland-Verville et al., 2015*; *Martial et al., 2017*; *Martial et al., 2018*; *Cassol et al., 2018*), we found a much higher rate of people stating that their experience was unpleasant. Although experiences with a cut-off score of at least 7 points on the GNDES were more often pleasant (53%) than experiences with a lower score (14%; $p < 0.0001$), almost half of all near-death experiences were labelled as stressful. It should be noted that (*Thonnard et al., 2013*; *Charland-Verville et al., 2015*; *Cassol et al., 2018*) included participants who had experienced near-death experiences in life-threatening situations only, i.e., excluding experiences made in non-life-threatening circumstances, which might have led to different frequencies of reported positive and negative emotions. In addition, however, we think the discrepancy between the present and many previous studies is likely because the GNDES addresses only pleasant feelings but does not include negative emotions, in contrast to our questionnaire (*Greyson, 1983*). Indeed, in line with our results, a recent study revealed a rather high proportion of negative near-death experiences (14%) (*Cassol et al., 2019*). Interestingly, and in accordance with our assumptions, that study also included near-death experiences of different etiologies and showed a higher proportion of suicidal attempts among people with negative near-death experiences (*Cassol et al., 2019*).

**Table 3  Participants claiming a near-death experience, analyzed according to Greyson Near-Death Experience Scale scores.** A score of ≥7 confirms the reported experience as a near-death experience.

| | All claimed NDE (*n* = 289) | Greyson NDE score <7 (*n* = 183) | Greyson NDE score ≥7 (*n* = 106) | *p*-value |
|---|---|---|---|---|
| **Greyson NDE score**—median (IQR) | 5 (3–8) | 4 (3–5) | 9 (8–14) | |
| Age—mean ± SD | 30.8 ± 11.1 | 30.0 ± 10.7 | 32.0 ± 11.6 | 0.14 |
| **Gender**—n (%) | | | | 0.34 |
| Female | 145 (50%) | 98 (54%) | 47 (44%) | |
| Male | 142 (49%) | 84 (46%) | 58 (55%) | |
| Other | 2 (0.7%) | 1 (0.5%) | 1 (0.9%) | |
| **REM intrusion**—n (%) | | | | **0.0003** |
| ≤ 2 criteria | 192 (66%) | 136 (74%) | 56 (53%) | |
| ≥ 3 criteria | 97 (34%) | 47 (26%) | 50 (47%) | |
| **Life-threatening event**—n (%) | | | | 0.55 |
| Yes | *158 (55%)* | *103 (56%)* | *55 (52%)* | |
| No | *131 (45%)* | *80 (44%)* | *51 (48%)* | |
| **Feelings associated with NDE \***—n (%) | (*n* = 230) | (*n* = 153) | (*n* = 77) | **<0.0001** |
| Unpleasant | 168 (73%) | 132 (86%) | 36 (47%) | |
| Pleasant | 62 (27%) | 21 (14%) | 41 (53%) | |

Notes.

IQR, interquartile range; n, number of participants; NDE, near-death experience(s); REM, rapid eye movements; SD, standard deviation.

Significant *p*-values are shown in bold script; *excluding participants reporting that their experience was neither pleasant nor unpleasant.

Again unlike previous studies (*Thonnard et al., 2013*; *Charland-Verville et al., 2015*; *Martial et al., 2017*; *Martial et al., 2018*; *Cassol et al., 2018*), we found that near-death experiences occurred equally likely in truly life-threatening situations and situations that only just felt so. This discrepancy probably results from the fact that we interviewed unprimed lay people from a large cross-sectional sample, whereas previous studies were retro- or prospectively performed in specific populations such as cardiac arrest survivors (*Van Lommel et al., 2001*; *Cassol et al., 2018*). This substantiates previous reasoning that near-death experiences are real experiences and not merely products of fantasy proneness (*Martial et al., 2018*): People with confirmed near-death experiences (i.e., GNDES score ≥7 points) did not perceive their situations as more dangerous than those whose experience did not qualify as a near-death experience (i.e., GNDES score ≤ 6 points), which argues against tendencies towards overdramatizing.

## Near-death experiences and REM sleep intrusion

Our central finding is that we confirmed a possible association of near-death experiences with REM sleep intrusion. Following multivariate analysis, REM sleep intrusion was the only factor that remained significantly correlated with near-death experiences (and indeed very much so: *p* = 0.0001). This finding corroborates and extends data from the previous study by Nelson and co-workers, in which the life-time prevalence of REM sleep intrusion in 55 humans with near-death experiences was compared with that in age- and sex-matched controls (*Nelson et al., 2006*). Sleep-related visual and auditory hallucinations and/or sleep paralysis assessed by a questionnaire like the one used in our study were substantially

**Table 4 Multivariate logistic regression and odds ratios for having a near-death experience (Greyson Near-Death Experience Scale ≥ 7).** To adjust for multiple testing, the alpha level was set to 0.01.

|  | OR (CI 95%) | *p*-value |
| --- | --- | --- |
| **Age** | 1.01 (0.99–1.04) | 0.35 |
| **Gender** | | |
| Female (reference) | 1.00 | |
| Male | 1.58 (0.93–2.69) | 0.09 |
| Other | 1.73 (0.05–59.89) | 0.74 |
| **Work** | | |
| Full-Time (reference) | 1.00 | |
| Part-Time | 0.47 (0.22–0.99) | 0.05 |
| Job seeking | 0.59 (0.16–1.93) | 0.40 |
| Not in paid work | 0.96 (0.35–2.58) | 0.94 |
| Student | 0.54 (0.25–1.15) | 0.11 |
| Other | 0.45 (0.15–1.3) | 0.15 |
| **Continent** | | |
| Americas (reference) | 1.00 | |
| Asia | 0 (0-NA) | 0.98 |
| Europe | 0.37 (0.12–1.05) | 0.06 |
| Oceania | 0.59 (0.05–5.79) | 0.65 |
| Other | 0.49 (0.08–2.92) | 0.43 |
| **REM intrusions** | | |
| ≤2 criteria (reference) | 1.00 | |
| ≥3 criteria | 2.85 (1.68–4.88) | **0.0001** |
| **Life-threatening event** | | |
| No (reference) | 1.00 | |
| Yes | 0.85 (0.5–1.43) | 0.53 |

**Notes.**

CI, confidence interval; OR, odds ratio; REM, rapid eye movements.
Significant *p*-values are shown in bold script.

more common in cases with near-death experiences. The authors suggested that under circumstances of peril, near-death experiences are more likely in people with a tendency towards REM sleep intrusion and that REM sleep intrusion might explain much, if not all, of the semiology of these experiences (*Nelson et al., 2006*). Indeed, as shown in Table 5, two participants from our study gave spontaneous reports of classic REM sleep disturbances (rather than reporting their near-death experience as requested) akin to those seen in people with narcolepsy (*Kondziella & Arlien-Soborg, 2006*; *Scammell, 2015*).

*Nelson et al. (2006)* based the hypothesis of REM intrusion being associated with near-death experiences on several "lines of evidence": REM intrusion during wakefulness occurs frequently in healthy people (from 1.2–32% for cataplexy to 24–28% for hypnagogic hallucinations (*Ohayon et al., 1999*; *Cheyne, Rueffer & Newby-Clark, 1999*)); REM intrusion is the hallmark of narcolepsy that shares semiological features with near-death experiences; and cardiorespiratory afferents may evoke REM intrusion by heightened vagal afferent activity (*Nelson et al., 2006*). Further, complex dream-like hallucinations like

**Table 5  Selected reports from participants with an experience that reached the threshold of ≥7 points on the Greyson NDE scale to qualify as a near-death experience.** Note that the last two comments describe instances that are highly suggestive of REM sleep disturbance, including visual hypnogogic hallucinations and sleep paralysis, rather than the near-death experience both participants reported to have had. Comments are edited for clarity and spelling.

- I was at the beach in Florida, I was 10–11. Suddenly, huge waves started pulling me further and further from the shore. As I was fighting, my life started flashing before me in my head. [ …] I felt like my soul was being pulled out of my body. I was floating and was [lifted in the air]. After a few moments I felt like I was in an enormous tunnel of darkness, and at its end there was the brightest white light I have ever seen. I remember that my dead relatives were at the gate, including my maternal grandmother. I don't remember what we talked about. But then I felt that I was sucked out of the tunnel and I fell, crashing into my body again. *Male, 28 years; near-drowning; Greyson NDE Scale score 10; fulfilling 3 of 4 criteria for REM sleep intrusion*

- I encountered a truly out-of-body experience where my eyesight and visual became incredibly abstract. For around an hour I had no sense of self or my surroundings. When my self-awareness returned, I became concerned that I was indeed dying or had died. I eventually became completely lucid, and still to this day I do not understand this experience. *Male, 46 years; drug intoxication; Greyson NDE scale 8; REM 3/4*

- I was very young when I almost drowned. I saw angels, and they were singing the most beautiful music I have ever heard. I was very upset when I was revived. *Female, 57 years; near-drowning; Greyson NDE scale 15; REM 1/4*

- During my first cardiorespiratory arrest I was aware of being outside my body. My partner saw me at the window, calling for help, but at this point I was not breathing. *Female, 35 years; critical illness/cardiac attack; Greyson NDE scale 25; REM 3/4*

- I felt like I just died, and I went to heaven. I heard voices, and I was sure I would not come back to my life. It was weird. I could not control my body. *Female, 37 years; childbirth; Greyson NDE scale 11; REM 3/4*

- It was a very pleasant experience: Intense white light, feelings of overwhelming love. I had a sense of not having done all the things I was meant to do. I heard a nurse repeatedly calling my name and telling me to breathe. I eventually took a breath. It was a very positive experience and has affected my whole life since in a very positive way. Female, 59 years; childbirth complication; *Greyson NDE scale* 15; REM 2/4

- I nearly drowned when I was around 8 years old. I felt total peace. Twenty years later I can still remember how I felt. It was an amazing feeling. *Female, 32 years; near-drowning; Greyson NDE scale 7; REM 2/4*

- I often see characters in my hallway or feel someone else's presence before going to sleep. *Male, 32 years; near-drowning; Greyson NDE scale 11, REM 4/4*

- Sometimes I wake at night, and I can't move. I see strange things, like spirits or demons at my door, and after a while I see them coming beside me. I can't move or talk, and they sit on my chest. It scares the hell out of me! I think that it is a dream, count to 3 and close my eyes. Sometimes this helps. *Female, 28 years; physical violence; NDE 20; REM 4/4*

those of REM sleep are well-described with lesions near the mesopontine paramedian reticular formation and the midbrain cerebral peduncles (i.e., peduncular hallucinations) (*Galetta & Prasad, 2017*), suggesting that dysfunction of the REM-inhibiting serotonergic dorsal raphe nuclei and the noradrenergic locus ceruleus may facilitate REM intrusion (*Hobson, McCarley & Wyzinski, 1975*; *Manford & Andermann, 1998*; *Kayama & Koyama, 2003*; *De Lecea, Carter & Adamantidis, 2012*; *Hasegawa et al., 2017*). Nelson and co-workers therefore hypothesized that people with near-death experiences may have an arousal system predisposing them to REM intrusion (*Nelson et al., 2006*).

The anonymous nature of our online survey enabled us to avoid the selection bias that was a major point of criticism (*Long & Janice Miner, 2007*) of the Nelson et al. study (*Nelson*

**Table 6  Selected reports from participants with an experience below the threshold of ≥7 points on the Greyson NDE scale.**  Comments are edited for clarity and spelling.

- I felt extreme fear and was certain I would die. At one point I suddenly stopped resting against what was certain to come, and instead a feeling of complete calm and acceptance came over me. I was fully in the moment and had no thoughts of anything else. When I was out of danger, I was shaken but the memory of the "good feeling" was clear. *Female, 50 years; psychological distress without organic disease; Greyson NDE Scale score 3; fulfilling 0 of 4 criteria for REM sleep intrusion*
- I hit the back of my head on a swimming pool springboard. I remember seeing myself outside of my body being helped, while I was unconscious. *Male, 32 years; concussion; Greyson NDE scale 4, REM 2/4*
- During a fire evacuation of an 18-story building, I apparently slept through and didn't evacuate. However, I experienced myself in peace, floating in the hallways and watching the other residents evacuate the building. Talking to them in the following days I could describe who passed me, and what they took with them or were wearing. *Male, 46 years; fire evacuation; Greyson NDE scale 6, REM 1/4*
- I lost a lot of blood during my last childbirth. I felt floaty and weird as if I was about to leave my body. I didn't feel like I was there anymore. *Female, 24 years; childbirth complication; Greyson NDE scale 5; REM 2/4*

*et al., 2006*). While the latter used a case-control approach, we evaluated an unselected sample of unprimed adult lay people. We carefully adjusted for confounding factors including age, gender, place of residence, employment status and perceived danger, and we found that the association of near-death experiences and evidence for REM intrusion remained statistically highly significant. Of note, online surveys limit the influence of psychological bias as compared to face-to-face interviews or telephone surveys because there is no incentive to please the investigator by inventing or exaggerating memories (*Peer et al., 2017*). (There was no monetary incentive to do so either, since we instructed participants that their reimbursement was the same irrespective of whether they would report a near-death experience or not). Also, we recruited a much larger sample size than what can be achieved with lab-based behavioral testing or case-control studies, including respondents from 35 countries, which strengthens the validity and generalizability of our results. It therefore appears that the association of near-death experiences and REM intrusion is real, although future work needs to address the relationship between the two states: Does a tendency to REM intrusion predispose to near-death experiences, as *Nelson et al. (2006)* argue, or could it be the other way round, i.e., do near-death experiences lead to a propensity for REM sleep intrusion (*Long & Janice Miner, 2007*)? Either way, we suggest that characterizing the precise biological mechanisms leading to REM intrusion into wakefulness might offer new insights into the physiology of near-death experiences.

## Limitations

Despite the advantages outlined above, online studies have limitations that should be acknowledged (*Woods et al., 2015*; *Peer et al., 2017*). First, complex clinical and ethical notions are impossible to fully implement in a survey form. For instance, almost a third of our participants claimed to have had two or more near-death experiences or near-death-like experiences, but we do not know if these participants only referred to one experience when completing the GNDES or several, which might have skewed GNDES scores in some cases. Second, although we assessed various demographic factors, there are many with potential importance such as religiosity (*Greyson, 2006*; *Chandradasa et*

*al., 2018*) that we did not record. (Although we did assess religiosity in a previous online survey using the same crowdsourcing platform and found that most participants have a secular background (*Kondziella, Cheung & Dutta, 2019*)). We did neither inquire about medical history (e.g., diagnoses of epilepsy, narcolepsy, sleep deprivation and migraine), medication or use of alcohol and illicit drugs. Third, for obvious (and ethical) reasons participants were informed about the topic of the survey, which could have led to a selection bias by evoking the interest of survey participants with near-death experiences (and hence overestimation of the true prevalence of such experiences). However, as stated above, the prevalence rate in our study is only slightly higher than that reported in earlier studies (*Knoblauch, Schmied & Schnettler, 2001*; *Perera & Padmasekara, 2005*; *Facco & Agrillo, 2012*), and the entire survey was finished during such a short time frame (i.e., the number of required participants was reached within 3.5 h) that word-of-mouth communication of the survey's topic appears very unlikely. Also, REM sleep intrusion was assessed using a self-reported questionnaire, the sensitivity and specificity of which remains undefined, and objective sleep testing might have yielded different results. However, sleep studies including polysomnography to confirm reports of REM intrusion would have been impractical in such a large multinational sample, and our questionnaire was identical to that from *Nelson et al. (2006)*, reflecting screening for REM sleep intrusion as performed by neurologists in clinical practice (*Kondziella & Arlien-Soborg, 2006*; *DelRosso, Chesson & Hoque, 2013*; *Scammell, 2015*; *Schneider & Mignot, 2017*). Lastly, it should be noted that we did not inquire about the frequency of REM intrusion or the temporality of near-death experiences and REM intrusion. Thus, we do not know which of them occurred first; and if there is a causal relationship, the direction remains unknown, i.e., REM sleep intrusion might predispose to or be induced by near-death experiences.

## CONCLUSIONS

In our study, the prevalence of near-death experiences was around 10%. Using a large intercontinental sample of lay people assessed by crowd-sourcing methods, we replicated findings from an earlier case-control study suggesting an association of near-death experiences with REM sleep intrusion (*Nelson et al., 2006*). Thus, whereas age, gender, place of residence, employment status and factual danger of the situation do not appear to influence the frequency with which near-death experiences occur, there seems to be a significant association with REM sleep intrusion. Although association is not causality, identifying the physiological mechanisms behind REM intrusion into wakefulness might advance our understanding of near-death experiences.

### Funding

This work was supported by grants of RH Forskningspulje (R143-A6132-B3632) to Dr. Daniel Kondziella and the Deutsche Forschungsgemeinschaft (DFG) DFG DR 323/5-1 and DFG DR 323/10-1, and FP7 no 602150 CENTER-TBI to Dr. Jens P. Dreier. The funders

had no role in study design, data collection and analysis, decision to publish, or preparation of the manuscript.

## Grant Disclosures

The following grant information was disclosed by the authors:
RH Forskningspulje: R143-A6132-B3632.
Deutsche Forschungsgemeinschaft (DFG): DFG DR 323/5-1, DFG DR 323/10-1.
FP7: 602150.

## Competing Interests

The authors declare there are no competing interests.

## Author Contributions

- Daniel Kondziella conceived and designed the experiments, performed the experiments, analyzed the data, contributed reagents/materials/analysis tools, prepared figures and/or tables, authored or reviewed drafts of the paper, approved the final draft.
- Jens P. Dreier analyzed the data, authored or reviewed drafts of the paper, approved the final draft.
- Markus Harboe Olsen analyzed the data, prepared figures and/or tables, authored or reviewed drafts of the paper, approved the final draft.

## Human Ethics

The following information was supplied relating to ethical approvals (i.e., approving body and any reference numbers):

The Ethics Committee of the Capital Region of Denmark waives approval for online surveys (Section 14 (1) of the Committee Act. 2).

## Data Availability

All anonymized raw study data are available in a Supplemental File. The data analysis plan is available at the Open Science Framework: https://osf.io/ykr3g.

## Supplemental Information

Supplemental information for this article can be found online at http://dx.doi.org/10.7717/peerj.7585#supplemental-information.

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
