# Peer review of "Prevalence of near-death experiences in people with and without REM sleep intrusion"

_PeerJ, doi:10.7717/peerj.7585_

## Round 0.1 · original submission · Major Revisions

This is an interesting study. Please revise it accordingly.

·

Basic reporting

Well constructed throughout, logical, good standard of English with appropriate supportive evidence.

Figures and tables relevant to the work. Figures highlighting some statements made by participants is fascinating and clearly upholds the claims made within the paper.

Line 68 and line 240 have unneeded hyphens between age- and sex-matched. These should be removed.

Experimental design

An interesting design using crowdsourcing, although limitations of this noted. Well described in terms of detail.

One question would be why you opened the survey for only 3.5 hours, as I read it (line294)? It would be beneficial to advise the reader why this decision was made as it appears to be a very short time frame to capture relevant data/allow participants to complete the survey especially given the global context and different time zones.

Unsure if using a survey dismisses incentive bias to please the investigator as with all surveys, the majority of participants have some engagement with the subject already but you have evidenced your comment.

Validity of the findings

Appropriate use of analysis tools bringing to the fore an obviously notable outcome in terms of statistics. Appears statistically sound.

Conclusion well drawn and consideration given to other outcomes or influences on the work.

Additional comments

A very interesting paper and was a pleasure to read.

I agree with regard to the positive nature of the GNDES and it would be good to be able to develop a new tool that offers the less positive experiences that some have demonstrated in your work.

Reviewer 2 ·

Basic reporting

Clear and correct english.
Some references can be added to strengthen the work
Tables and figures are acceptable

Experimental design

Ethical standards are duly fulfilled, however, technical standards are poor, since the information is obtained through an on-line survey.

Validity of the findings

Results must be handled as speculation and this must be identified as such

Additional comments

General Aspects
The present study depicts relevant information regarding the propensity of REM sleep intrusion in association to near-death experiences. However, the omission of arousal disorders weakens the strength of the work and of the data presented. Another limitation is that frequency and temporality of near-death experiences and REM intrusion is unknown, it is not known which of them occurred for the first time, so the results describe both phenomena barely. Authors should clarify if REM sleep intrusion is properly classified by the AASM. It is unclear if the intrusion of REM is different from sleep paralysis and if this phenomenon occurs only accompanied by narcolepsy, sleep paralysis may occur in non-narcoleptic subjects. Statistical analyzes have been well performed and are adequate. It would be interesting to know if subjects after the near-death experience present neurological damage. Sociodemographic data is the most interesting part of the work. Authors should have included in their instrument / scale / questionnaire more sleep parameters items, such as: diagnosed narcolepsy, epilepsy, sleep debt, insomnia, voluntary sleep deprivation, usual amounts of sleep, napping or not, apnea , etc. This would have strengthened its methodology.Authors should add information that supports or strengthens their work.
There are some aspects of the study which need to be revised.
Introduction
Lines 60 to 62, REM sleep definition must have a reference
Line 62, REM as a whole state cannot intrude into wakefulness, they are two different states, however, some characteristics of REM sleep can be present during awakening/wakefulness.
Results
Lines 161 to 164, Why were the truly life threatening and the not truly life-threatening not analyzed in independent groups?
Line 168, authors can consider that substance abuse (n = 33) could be removed from the sample. In these subjects the near-death experience must have been accompanied by altered states of consciousness. What was the rationale to include them?
Line 189, What is the evidence for REM intrusion? Is the scale score?
Discussion
Lines 255 to 260, consider to revise and improve this lines. References are correct but authors do not present the information correctly.
Line 256, consider to change “release”
It would have been important to know and describe the actual use of stimulants, alcohol and drugs of the survey participants.
Line 296 to 297, I ask the authors what are these different results?
Line 297 to 298, “sleep studies…….would have been impractical”. Authors should be more careful to make that statement, since the data that you present is obtained from subjective reports. Polysomnography is currently the gold standard for assessing sleep and it would have been desirable for the present or future studies,and a statement regarding this subject should be addressed in the limitations section.
I would recommend that authors can review the following references and if possible that some of them could be added to the manuscript:
Serotonin neurons in the dorsal raphe mediate the anticataplectic action of orexin neurons by reducing amygdala activity. Hasegawa E, Maejima T, Yoshida T, Masseck OA, Herlitze S, Yoshioka M, Sakurai T, Mieda M. Proc Natl Acad Sci U S A. 2017 Apr 25;114(17):E3526-E3535. doi: 10.1073/pnas.1614552114.
A clinician's guide to recurrent isolated sleep paralysis. Sharpless BA. Neuropsychiatr Dis Treat. 2016, 19;12:1761-7. Review
Sleep Paralysis, "The Ghostly Bedroom Intruder" and Out-of-Body Experiences: The Role of Mirror Neurons. Jalal B, Ramachandran VS. Front Hum Neurosci. 2017 Feb 28;11:92. doi: 10.3389/fnhum.2017.00092.
Sleep paralysis and "the bedroom intruder": the role of the right superior parietal, phantom pain and body image projection. Jalal B, Ramachandran VS. Med Hypotheses. 2014 Dec;83(6):755-7. doi: 10.1016/j.mehy.2014.10.002.
Sleep-related epileptic behaviors and non-REM-related parasomnias: Insights from stereo-EEG. Gibbs SA, Proserpio P, Terzaghi M, Pigorini A, Sarasso S, Lo Russo G, Tassi L, Nobili L. Sleep Med Rev. 2016 Feb;25:4-20. doi: 10.1016/j.smrv.2015.05.002.
Frequent lucid dreaming associated with increased functional connectivity between frontopolar cortex and temporoparietal association areas. Baird B, Castelnovo A, Gosseries O, Tononi G. Sci Rep. 2018 Dec 12;8(1):17798. doi: 10.1038/s41598-018-36190-w.
Neurobiological mechanisms for the regulation of mammalian sleep-wake behavior: reinterpretation of historical evidence and inclusion of contemporary cellular and molecular evidence. Datta S, Maclean RR. Neurosci Biobehav Rev. 2007;31(5):775-824. Epub 2007 Mar 12. Review.

Reviewer 3 ·

Basic reporting

The authors in this study investigated: (i) the frequency of near-death experiences and REM sleep intrusions collected from an online crowdsourcing service; (ii) the correlation between people who report a near-death experience with the frequency of REM sleep intrusions. They found that from a large sample of 35 countries the prevalence of near-death experiences was 10%. Additionally, they confirmed the association of near-death experiences with REM sleep intrusion.

Results obtained support the main hypothesis of the study.

The occurrence of REM sleep intrusion is very low in the general population, although the vast majority of cases go unrecognized and its prevalence change in relation to age, with the use of antidepressant medications or in the setting of narcolepsy.
However, the authors discussed that " REM intrusion during wakefulness occurs frequently in healthy people" (see discussion section line 250). Please, make a citation on this sentence and also indicated in the discussion the percentage of REM sleep intrusion in the general population compared to your population.


I would ask whether it is possible to define in your population among people who had REM sleep intrusion the kind of medications (i.e antidepressive) used. Additionally, may you determine among people who had REM sleep intrusion people who may suffer from narcolepsy? Please indicate if it is possible this data in the results or makes comments on this.

Additionally, you addressed your hypothesis at the beginning of the section Materials and methods. Please, I would add the hypothesis and research question in the introduction at the of the paragraph.

Experimental design

Research question well defined
However, I do not totally agree how the REM sleep intrusion has been determined.

Validity of the findings

The study confirms the data already published from the previous study (Nelson at al 2006), in which the authors correlated the NDE with REM sleep intrusion.
However, this study was criticized for several regions including the control group used.
I believe that in this study the association between the NDE and REM sleep intrusion is still speculative since the way how REM sleep intrusion has been determined is limited. To this, I would suggest to the authors to add the word "we speculate..." throughout the manuscript

Reviewer 4 ·

Basic reporting

The paper is of interest and adds a relevant piece to the near-death experience (NDE) literature. It assesses the prevalence of NDEs in people presenting or not REM sleep intrusions. I am not a native English speaker, therefore, I cannot judge adequately about the quality of the writing. I will however note that the manuscript seems rather well written, clear and easy to read. Moreover, authors have made all the appropriate raw data available.

The introduction is brief, giving some context for both the study of NDEs and REM intrusion. However, it would benefit the reader to hear more about studies that have examined the incidence and prevalence of such experiences, although some of them were subject to methodological biases. Some of these studies have been mentioned by the authors in the Methods and Discussion, but more might be said about this in the Introduction. In this respect, see the following articles:
-Knoblauch H., Schmied I., Schnettler B. 2001. Different Kinds of Near-Death Experience: A Report on a Survey of Near-Death Experiences in Germany. Journal of Near-Death Studies 20:15–29.
-van Lommel P., van Wees R., Meyers V., Elfferich I. 2001. Near-death experience in survivors of cardiac arrest: a prospective study in the Netherlands. The Lancet 358:2039–2045. DOI:10.1016/S0140-6736(01)07100-8.
-Perera M, Padmasekara G BJ. 2005. Prevalence of Near-Death Experiences in Australia. Journal of Near-Death Studies 24:109–116.

Experimental design

Overall, I believe that this original and meaningful research fits the aims and scope of the journal. Nonetheless, some points need clarification:

>Authors write that they also inquired about unpleasant experiences. Could they detail/specify the question(s) they asked in the methods section?

>Authors claim that they have been inspired by the study by Nelson et al. (2006) which suffered a selection bias, and that the anonymous nature of their online survey enabled the avoidance of a selection bias. However, it seems to me that the title of the survey, which places emphasis on NDEs, could have introduced a selection bias by overestimating the number of participants having experienced a NDE. I would suggest that authors comment on this point.

Validity of the findings

Overall, data are relevant and well analyzed. Nevertheless, t is not clear to me why the alpha level was set at 0.01 in the case of a multivariate logistic regression. Therefore, I wonder whether the authors could comment on that point?

Additional comments

Overall, authors need to check the references in the text which are not always written in a coherent manner.

ASTRACT:
There is an extra space at the end of the Methods part.

MATERIALS AND METHODS:
>Line 110: authors state that the survey ended whenever participants declared not having experienced a NDE. Besides, in Table 1, NDEs are defined as events occurring in life-threatening situations or in situations that just felt so. Therefore, it is not clear to me how they ended up with a sample of experiencers that also included individuals that have experienced a NDE during meditation or prayer. Could authors comment on this point?

RESULTS:
>Line 184: some participants stated having had two or three NDEs or NDEs-like, but can authors ensure that participants only referred to one experience when completing the GNDES? That they have not indicated all the characteristics they have experienced during their various NDEs (which could, in this case, lead to false positives on the GNDES)?
>Line 190: I recommend using the acronym GNDES, as it was previously done in the manuscript.

DISCUSSION:
>Line 218:
-I would suggest removing the p-value from this part of the manuscript;
-On the same line, authors mention that almost half of all experiencers labeled their experience as “stressful”: was this determined on the basis of the written narratives or by using the additional close-ended question about the valence of the experience?
>It could be interesting to cite the following recent article in which a proportion of 14% of negative NDEs was identified. Interestingly, and in accordance with authors’ assumption, this study also included NDEs of different etiologies (and not only cardiac arrests) and showed a higher proportion of suicidal attempts among negative NDEs:
Cassol, H., Martial, C., Annen, J., Martens, G., Charland-Verville, V., Majerus, S., & Laureys, S. (2019). A systematic analysis of distressing near-death experience accounts. Memory, 1–8. https://doi.org/10.1080/09658211.2019.1626438
>Line 222: authors state that NDEs occurred equally in life-threatening as well as non-life-threatening situations, supposedly unlike other studies, nevertheless, the studies that they cite are not adequate to illustrate this point. Indeed, Thonnard et al., 2013, Charland-Verville et al., 2015 and Cassol et al., 2018 voluntarily chose to screen participants who had experienced NDEs in life-threatening situations. These participants come from the database of the Coma Science Group which also comprises NDEs experienced after non-life-threatening situations (see Charland-Verville et al., 2014). Regarding Martial et al., 2017, authors do not specify the context in which their participants experienced their NDE.

TABLES:
>In Table 3, I recommend to use a dash instead of a semicolon when reported the IQR.
>In the legends of Table 4 and Table 5, authors should remove “n – number of participants”.

---

## Round 0.2 · Minor Revisions

Most issues have been addressed by the authors. One reviewer still has a minor comment for the paper. Please kindly consider and revise it. Before re-submission, please carefully check language of the paper again.

·

Basic reporting

No further comment

Experimental design

No further comment

Validity of the findings

No further comment

Additional comments

Thank you for the amendments made. This reads well and is a very interesting subject.

Reviewer 2 ·

Basic reporting

no comment

Experimental design

no comment

Validity of the findings

no comment

Additional comments

Authors really improved the manuscript. It should be noted that changes strengthen the work. The work is now clearer and contains more information about near-death experiences and iits relation with REM sleep intrusion. Details clarified in the online survey strengthen the work. Improvements at the same time strengthened the results.

Reviewer 3 ·

Basic reporting

Thanks the authors for fully and clearly replay to the answers.

Experimental design

Ethical standards are duly fulfilled. However, there are limitations given by an on-line survey.

Validity of the findings

The erects support the main hypothesis

Reviewer 4 ·

Basic reporting

All comments have been addressed.

Experimental design

All comments have been addressed.

Validity of the findings

All comments have been addressed

Additional comments

Most issues have been addressed; however, it seems to me that the authors may not have quite seized one of my concerns. Yes, authors have stated several times in the manuscript that the near-death experience phenomenology may also occur during non-life threatening situations. However, in their survey, participants were asked the following question: "(…)near-death experiences are exceptional experiences that you may have when you are dying or feel as if you were dying. Have you ever had such a near-death experience – either during a true lifethreatening event or an event that just felt so?". Therefore, even though participants may not have been in serious danger, they should have at least felt like it in order to pursue the survey (in case of negative answer, the survey ended there). Knowing that, it is surprising to note that the final sample also includes participants who have experienced a NDE phenomenology during meditation or prayer, which in principal are not life-threatening contexts, whether objectively or subjectively. Could authors comment on this point? Did they gather information that could inform on such a result?

---

## Round 0.3 · accepted · Accept

I think this revised version can be accepted now.